# Node of Ranvier length as a potential regulator of myelinated axon conduction speed

I Lorena Arancibia-Cárcamo[1†], Marc C Ford[1†], Lee Cossell[1†], Kinji Ishida[2], Koujiro Tohyama[2,3]*, David Attwell[1]*

[1]Department of Neuroscience, Physiology and Pharmacology, University College London, London, United Kingdom; [2]The Center for Electron Microscopy and Bio-Imaging Research, Iwate Medical University, Morioka, Japan; [3]Department of Physiology, School of Dentistry, Iwate Medical University, Morioka, Japan

**Abstract** Myelination speeds conduction of the nerve impulse, enhancing cognitive power. Changes of white matter structure contribute to learning, and are often assumed to reflect an altered number of myelin wraps. We now show that, in rat optic nerve and cerebral cortical axons, the node of Ranvier length varies over a 4.4-fold and 8.7-fold range respectively and that variation of the node length is much less along axons than between axons. Modelling predicts that these node length differences will alter conduction speed by ~20%, similar to the changes produced by altering the number of myelin wraps or the internode length. For a given change of conduction speed, the membrane area change needed at the node is >270-fold less than that needed in the myelin sheath. Thus, axon-specific adjustment of node of Ranvier length is potentially an energy-efficient and rapid mechanism for tuning the arrival time of information in the CNS.

*For correspondence: tohyama@coral.plala.or.jp (KT); d.attwell@ucl.ac.uk (DA)

[†]These authors contributed equally to this work

**Competing interests:** The authors declare that no competing interests exist.

## Introduction

During development or learning, an adjustment of myelin thickness or internode length may be used to tune the conduction speed of myelinated axons (*Fields, 2008*; *Tomassy et al., 2014*; *Ullén, 2009*; *Kimura and Itami, 2009*). This can promote synchronous neuronal firing (*Lang and Rosenbluth, 2003*; *Sugihara et al., 1993*), make impulse propagation time less dependent on the spatial trajectory of the axon transmitting information between areas (*Salami et al., 2003*), or adjust propagation delays to mediate sound localization (*Carr and Konishi, 1990*; *McAlpine and Grothe, 2003*; *Seidl et al., 2010*; *Ford et al., 2015*). Magnetic resonance imaging of humans and cellular studies of rodents suggest that myelination can increase while learning motor tasks such as piano playing (*Bengtsson et al., 2005*), juggling (*Scholz et al., 2009*), reaching (*Sampaio-Baptista et al., 2013*) and running (*McKenzie et al., 2014*).

Although most interest has focussed on the effect of changes of myelin thickness or internode length, the node of Ranvier is another potential determinant of action potential conduction speed. Increasing the length of the node will increase the node capacitance and the axial resistance for current flow into the internode, which will both decrease conduction speed. On the other hand, a greater length could increase the number of voltage-gated $Na^+$ channels at the node (if the channel density is constant), which may increase conduction speed. Given the potential influence of node length on conduction speed, we quantified heterogeneity of the length of the node of Ranvier in the white matter of the rat optic nerve and corpus callosum, and in the grey matter of rat cerebral cortex. Computer modelling was then used to explore the effects on conduction speed of the range of node lengths observed.

**eLife digest** Information is transmitted around the nervous system as electrical signals passing along nerve cells. A fatty substance called myelin, which is wrapped around the nerve cells, increases the speed with which the signals travel along the nerve cells. This allows us to think and move faster than we would otherwise be able to do.

The electrical signals start at small "nodes" between areas of myelin wrapping. Originally it was thought that we learn things mainly as a result of changes in the strength of connections between nerve cells, but recently it has been proposed that changes in myelin wrapping could also contribute to learning.

Arancibia-Cárcamo, Ford, Cossell et al. investigated how much node structure varies in rat nerve cells, and whether differences in the length of nodes can fine-tune the activity of the nervous system. The experiments show that rat nerve cells do indeed have nodes with a range of different lengths. Calculations show that this could result in electrical signals moving at different speeds through different nerve cells.

These findings raise the possibility that nerve cells actively alter the length of their nodes in order to alter their signal speed. The next step is to try to show experimentally that this happens during learning in animals.

## Results

### Variation of node length in the optic nerve and cortex

We first measured the length of nodes of Ranvier and axon diameter in adult rat optic nerve using both confocal and serial electron microscopy (*Figure 1*, *Figure 1—figure supplement 1*). Using electron microscopy (EM) we found the mean node length was 1.08 ± 0.02 µm (mean ± s.e.m., n = 46 nodes). Node lengths varied 2-fold, from 0.7 to 1.4 µm with a standard deviation of 0.15 µm (*Figure 1—figure supplement 1*). Node lengths were not significantly correlated with axon diameter at the node, which had a mean value of 0.80 ± 0.03 µm (standard deviation 0.19 µm; *Figure 1—figure supplement 1*, the slope of the regression line is not significantly different from zero, p=0.46). Thus, the different node lengths observed did not simply reflect axons being of different sizes.

In contrast to EM, confocal microscopy allowed us to measure a greater number of nodes in different parts of the CNS. In addition, given that oligodendrogenesis and myelination continues well into adulthood (*Dimou et al., 2008*; *Young et al., 2013*), confocal microscopy of antibody-labelled nodes allowed us to distinguish developing nodes from mature nodes, ensuring that any differences in size observed were not due to different developmental stages. Mature nodes of Ranvier were identified from their $Na_V1.6$ staining (a marker of mature nodes: *Boiko et al., 2001*; *Kaplan et al., 2001*) flanked by Caspr-labelled paranodes (*Figure 1A*), and node length was measured from the Caspr intensity profile across the node (*Figure 1B*, see Materials and methods). We first assessed whether, using confocal microscopy (*Figure 1C*), we could observe variability of node lengths in the rat optic nerve similar to that found when using EM. We again found node length variability (*Figure 1C,E,F,H*), the mean node length was not significantly different (p=0.06) from that obtained with EM (1.02 ± 0.02 µm, standard deviation 0.29 µm, n = 164, *Figure 1E*), and there was again no significant dependence on either node diameter (*Figure 1H*, p=0.14) or axon diameter at the paranode (measured as the diameter of the Caspr labelling: p=0.89). However, the node length range was slightly broader, covering a 4.4-fold range (*Figure 1F,H*), perhaps due to the ~4 fold greater number of nodes quantified.

In the grey matter of the adult cerebral cortex (layer V of motor cortex) an even larger, 8.7-fold, range of node lengths was observed (*Figure 1D,E,G,H*), from 0.43 to 3.72 µm (mean value ± s.e.m 1.50 ± 0.05, standard deviation 0.58 µm, n = 158). Again there was no significant dependence on node diameter (*Figure 1H*, p=0.42), the mean value of which was 0.64 ± 0.01 µm (n = 158, standard deviation 0.14 µm), or on axon diameter at the paranode (p=0.98).

Thus, variability of node length is a general feature of myelinated axons. The greater variability of node length observed in the cortex than in the optic nerve has parallels with the far greater

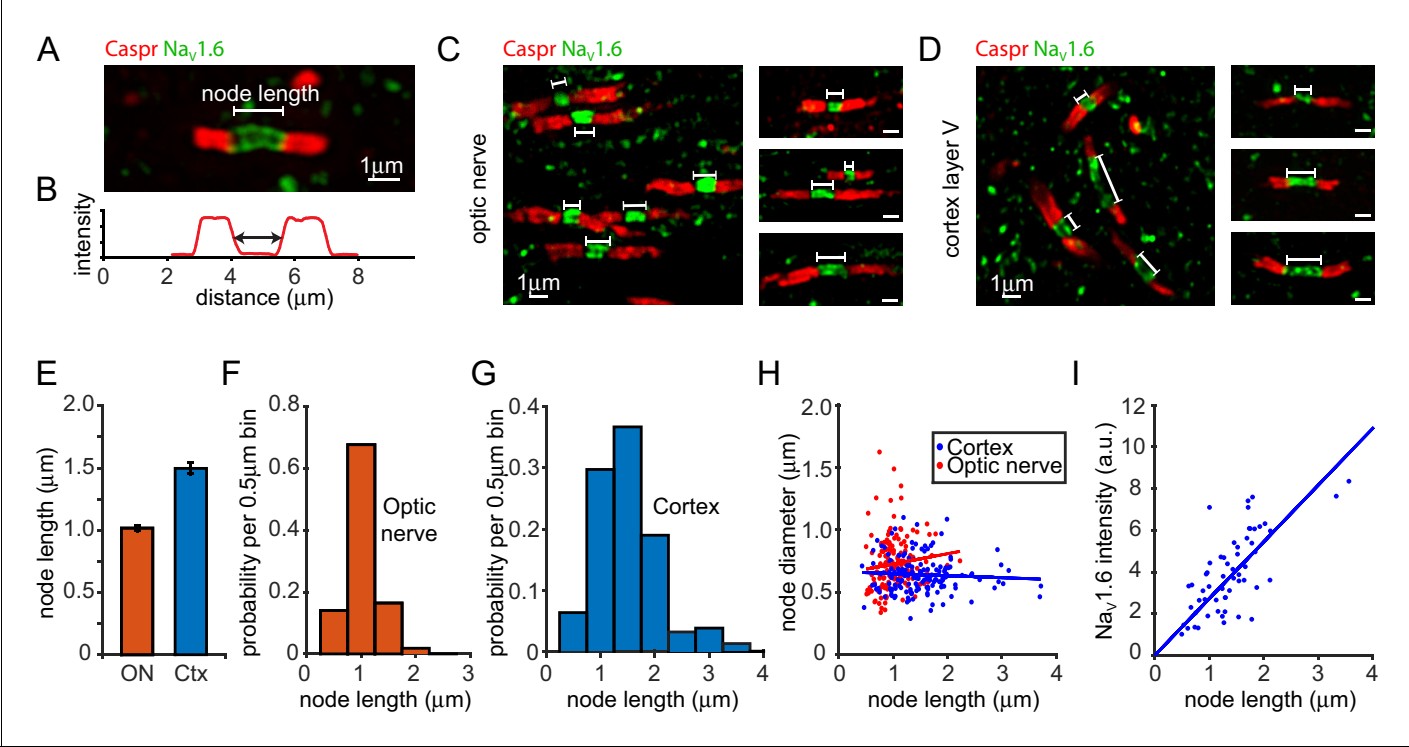

**Figure 1.** Heterogeneity of Ranvier node lengths in the optic nerve and cerebral cortex. (**A**) Confocal image of a single optic nerve node of Ranvier showing the node labelled with antibody to $Na_V1.6$ (green) and paranodes labelled for Caspr (red). (**B**) Intensity profile of Caspr staining for the node in A. Node length was measured as the distance between the half maximum intensity for each paranode. (**C, D**) Confocal images of nodes in the optic nerve (**C**) and layer V of the cortex (**D**) highlighting the different range of node of Ranvier sizes in these areas. (**E**) Mean ± s.e.m. of the node lengths measured in optic nerve (ON, red) and layer V of the cortex (blue). (**F, G**) Distribution of node lengths from data in E for optic nerve (**F**) and cortex (**G**). (**H**) Node diameter as a function of node length in the optic nerve (red) and in grey matter of the cerebral cortex (blue). Slope of regression lines in H are not significantly different from zero (p=0.14 (optic nerve) and p=0.42 (cortex)). (**I**) $Na_V1.6$ immunolabel intensity (summed over each node) as a function of node length in cerebral cortex (each point is one node). $Na_V1.6$ labelling is correlated with node length (slope of regression line is significantly greater than zero, $p=1.2\times10^{-15}$).

The following figure supplement is available for figure 1:

**Figure supplement 1.** Heterogeneity of Ranvier node lengths in the optic nerve.

variability of myelination seen in the adult cortex (*Tomassy et al., 2014*) and could reflect tuning of individual axon conduction speeds to meet information processing needs.

To investigate whether this node length variability was accompanied by a similar variability in sodium channel number, we summed the intensity of $Na_V1.6$ staining over individual nodes in cortex and compared this with node length (*Figure 1I*). We found that there is a linear correlation between summed $Na_V1.6$ staining and node length ($p=1.2\times10^{-15}$) indicating that the number of sodium channels at nodes is approximately proportional to node length (cf. *Rios et al., 2003*), consistent with larger nodes being generated by the insertion of more $Na_V$-containing membrane. However, at any given sodium channel labelling intensity there was still a large variation in node lengths, suggesting that sodium channel density is not absolutely constant, and that it may be possible to vary node length in a manner independent of sodium channel number.

## Node lengths vary far more between axons than along axons

If adjustment of node length is used to tune conduction speed, one might expect all the nodes along one axon to have similar lengths (e.g. all long or all short). To assess whether the variability in node lengths mainly occurs along axons or between axons we iontophoretically injected a fluorescent dye into the cortex of adult rats (see Materials and methods). This was taken up into axons and

diffused along them, allowing us to measure the lengths of up to 13 successive nodes (mean 6.7 ± 0.8 nodes in 18 axons) along single fluorescently labelled axons in the corpus callosum (*Figure 2A*). Remarkably, along individual axons (*Figure 2B,C*), the distribution of node lengths was much narrower than that observed over all callosal axons examined (*Figure 2C*), with a 48.8% ± 3.5% lower coefficient of variation (s.d./mean, *Figure 2D*, p=1.1×10$^{-10}$, one sample t-test, n = 18 axons). Importantly, even when excluding axons with mean node lengths >2.25 μm (which were found less frequently, as predicted by the distribution in *Figure 1G–H*), the coefficient of variation for single axons was 38.8% ± 4.7% lower than that observed over all axons (p=6.6×10$^{-7}$). Node length was not correlated with internode length (p=0.1, *Figure 2E*).

Thus, node lengths are similar along axons but differ significantly between axons. This raises the possibility that individual axons consistently adjust their node length to tune conduction speed.

## Predicted effects of node length variation on conduction speed

To examine the consequences of nodes of Ranvier having different lengths, we simulated action potential propagation in optic nerve and cortical grey matter myelinated axons, as described in the Materials and methods. The differential equations of the model were derived and solved as in *Halter and Clark (1991)*. Details of the parameters used are summarised in *Table 1*. The conduction speeds predicted for the mean node lengths observed (2.95 m/s for the optic nerve and 2.61 m/s for the cortex) were within the range of values observed experimentally in the adult rat optic nerve (2.5–15 m/s: *Foster et al. (1982)*; *Sefton and Swinburn (1964)*; *Sjöström et al. (1985)*) and for different classes of rat cortical grey matter output axons (1.8–5.9 m/s: *Kelly et al., 2001*).

Our data suggest a positive correlation between the number of Na$_V$1.6 channels and node length (indicating a fixed channel density), but also raise the possibility of node length varying in a manner independent of channel number (*Figure 1I*). We therefore modelled two extreme situations, for both the optic nerve and the cortical axons studied: either the density of nodal ion channels was assumed to be constant (so the number of ion channels increases in proportion to node length), or the number of ion channels at the node was held constant at the values assumed for the mean node length observed (so the density of channels varies inversely with node length).

*Figure 3A and B* show that, when the number of channels was held constant at each node, the predicted conduction speed falls with increasing node length (dashed curves). This occurs for two reasons: the increase in node length increases the nodal capacitance (so each node takes longer to charge), and the intracellular axial resistance to current flow from the node into the internode is increased. The changes in conduction speed for the optic nerve are shown in *Figure 3A* (the range of measured node lengths is shown for comparison). Increasing the node length from its mean value of 1.02 μm to the largest value observed (2.2 μm) is predicted to decrease the conduction speed by 6.5%, while decreasing the node length to the smallest value measured (0.5 μm) increases the speed by 3.2% (giving a speed that is 10.3% larger than at a length of 2.2 μm). For cortical axons (*Figure 3B*) the predicted changes are larger, partly because, with a 1.5-fold longer node and a 1.7-fold shorter internode length, the nodal membrane contributes a larger fraction of the total membrane capacitance (14% in cortical axons versus 8% in optic nerve axons). The node length variation observed in rat cortex (*Figure 1G–H* and 0.43 μm to 3.7 μm) results in conduction speeds that are 11.6% slower (for the longest node) and 7% faster (for the shortest node) than the speed for the mean node length of 1.5 μm. Thus, altering node length from 3.7 to 0.43 μm would increase the speed by approximately 21%.

When the nodal ion channel density is kept constant another factor affects the predicted conduction speed, in addition to the change of capacitance and axial resistance at the node: as the node length is decreased the reduction in the number of ion channels present leads to a decrease of conduction speed. Consequently, the plot of speed against node length shows a maximum (solid curves in *Figure 3A–B*: note that, above this maximum, increasing node length decreases speed for the reasons stated above, despite the increase in number of sodium channels at the node). Decreasing the optic nerve node length from its mean value of 1.02 μm to the lowest length observed (0.5 μm) is predicted to decrease conduction speed by 14.1%, while increasing node length to the value generating the maximum conduction speed (1.7 μm) increases the speed by 3.3% (*Figure 3A*), to a value that is 20.2% higher than at a length of 0.5 μm. Similarly, for the cortex, decreasing the node length from the mean value of 1.5 μm to the smallest observed value of 0.43 μm decreases the conduction

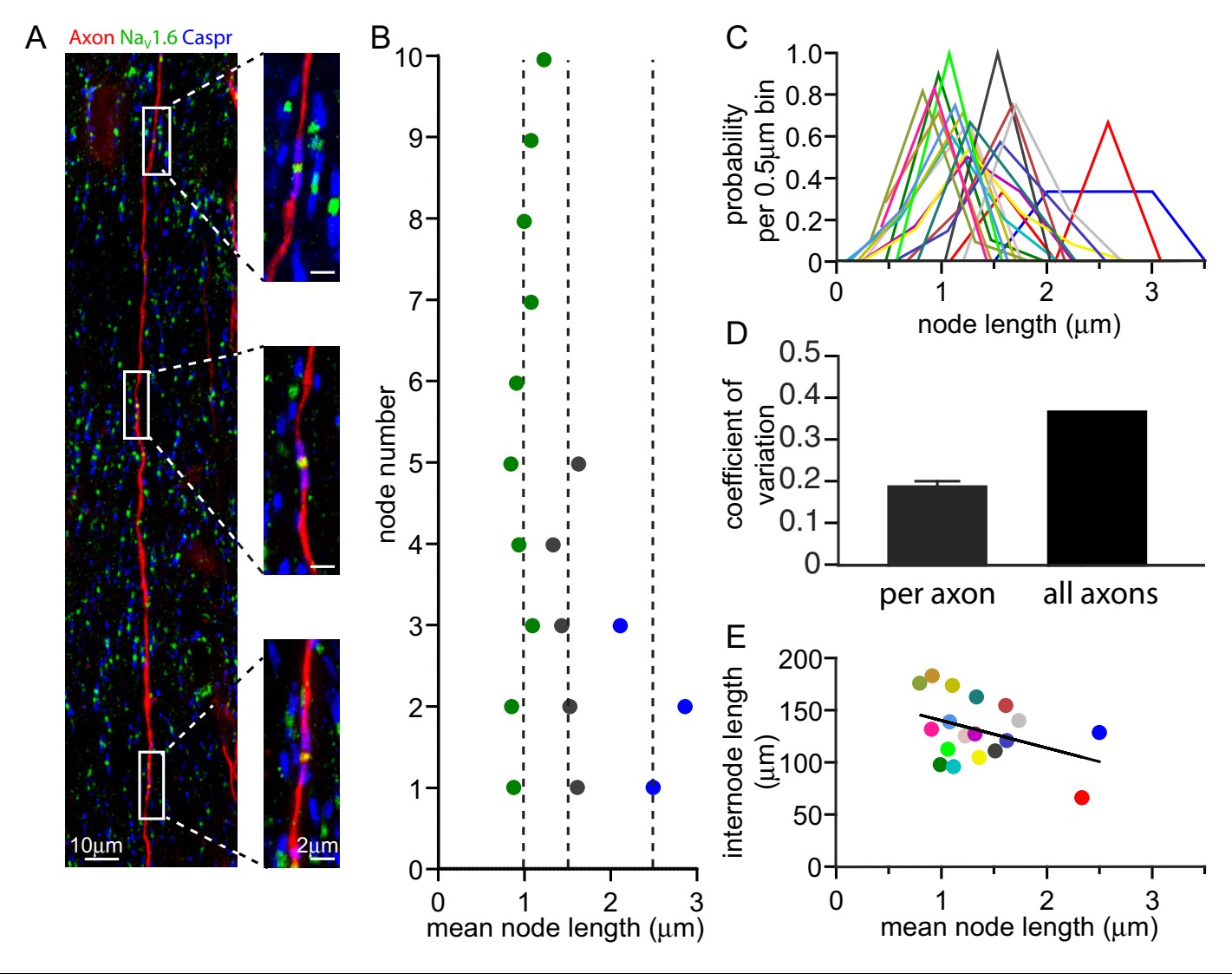

**Figure 2.** Node of Ranvier lengths are correlated along an axon. (A) Composite confocal image of a single axon in the corpus callosum iontophoretically labelled with tetramethylrhodamine dextran (red). Three consecutive nodes of Ranvier are highlighted and shown in high resolution images. Nodes of Ranvier are identified as Na$_V$1.6 positive clusters (green) flanked by Caspr positive paranodes (blue). (B) Successive node lengths along three example axons with different mean node lengths. The mean node length for each axon is plotted as a dashed line. (C) Distributions of node lengths of 18 individual axons (in 0.5 μm bins, centered on the median node length for each axon) show that the variability along each axon is much less than the variability between axons. (D) Mean coefficient of variation for node lengths along 18 individual axons and the overall coefficient of variation for all axons examined. (E) Mean internode length for each axon plotted against the mean node length for that axon. Each axon is represented by a different colour, and that colour is maintained for panels B, C and E. Regression line slope is not significantly different from zero (p=0.1).

velocity by 19.6% while increasing the length slightly to 1.7 μm increases the speed by 0.2% to a value 24.6% larger than at a node length of 0.43 μm (**Figure 3B**).

Although the node length is similar at successive nodes along individual axons (**Figure 2B**), some variation does exist (with a mean standard deviation of 0.25 ± 0.06 μm in 18 axons), so we assessed the effect of this by simulating a situation where alternate nodes had a length 0.25 μm shorter and 0.25 μm longer than the mean node length. For example, for a mean node length of 1.0 μm, alternate nodes had lengths of 0.75 μm and 1.25 μm. When the number of channels was held constant at each node, the approximately linear relationship between conduction velocity and node length

**Table 1.** Electrical and geometrical parameters of the models.

| Parameter | Symbol | Value | Units |
|---|---|---|---|
| Nodal Na$^+$ conductance[*] | $g_{Na}$ | 3000 | mS/cm$^2$ |
| Nodal K$^+$ conductance[*] | $g_{Ks}$ | 80 | mS/cm$^2$ |
| Nodal persistent Na$^+$ conductance[*] | $g_{Nap}$ | 5 | mS/cm$^2$ |
| Leakage conductance<br>Node[*]<br>Internode | $g_L$ | <br>80<br>0.1 | <br>mS/cm$^2$<br>mS/cm$^2$ |
| Myelin membrane conductance | $g_{my}$ | 1.0 | mS/cm$^2$ |
| Axon membrane capacitance[†]<br>Node<br>Internode | $c_{ax}$ | <br>0.9<br>0.9 | <br>µF/cm$^2$<br>µF/cm$^2$ |
| Myelin membrane capacitance[†,‡] | $c_{my}$ | 0.9 | µF/cm$^2$ |
| Axoplasmic resistivity | $\rho_{ax}$ | 70 | Ω.cm |
| Periaxonal resistivity | $\rho_P$ | 70 | Ω.cm |
| Resting potential | $E_r$ | −82 | mV |
| Leakage potential | $E_{Lk}$ | −83.38 | mV |
| Na$^+$ reversal potential | $E_{Na}$ | 50 | mV |
| K$^+$ reversal potential | $E_K$ | −84 | mV |
| Node diameter<br>Optic nerve<br>Cortex | | <br>0.73<br>0.64 | <br>µm<br>µm |
| Node length<br>Optic nerve<br>Cortex | | <br>1.02<br>1.50 | <br>µm<br>µm |
| Paranode length<br>Optic nerve<br>Cortex | | <br>2.11<br>1.90 | <br>µm<br>µm |
| Paranodal effective periaxonal space<br>Optic nerve<br>Cortex | | <br>0.0077<br>0.0123 | <br>nm<br>nm |
| Internodal axon diameter<br>Optic nerve<br>Cortex | | <br>0.82<br>0.73 | <br>µm<br>µm |
| Internodal periaxonal space | | 15 | nm |
| G ratio<br>Optic nerve<br>Cortex | | <br>0.78<br>0.81 | |
| Number of myelin wraps<br>Optic nerve<br>Cortex | | <br>7<br>5 | |
| Internode length<br>Optic nerve<br>Cortex | | <br>139.26<br>81.7 | <br>µm<br>µm |

[*]Values for standard node length: 1.02 µm in optic nerve, 1.50 µm in cortex; these are constant for simulations with fixed nodal conductance density, but scaled inversely with node length for simulations where number of nodal channels is kept constant.

[†]Membrane capacitance values are from **Gentet et al. (2000)**.

[‡]Figures are per myelin membrane. There are two membranes per myelin lamella.

(**Figure 3A,B**) resulted in a slightly faster propagation through shorter nodes and a slightly slower propagation through longer nodes, the effects of which cancel out with no overall effect on conduction velocity. When the nodal ion channel density was kept constant, the concave-downwards dependence of speed on node length in **Figure 3A–B** resulted in marginally slower conduction

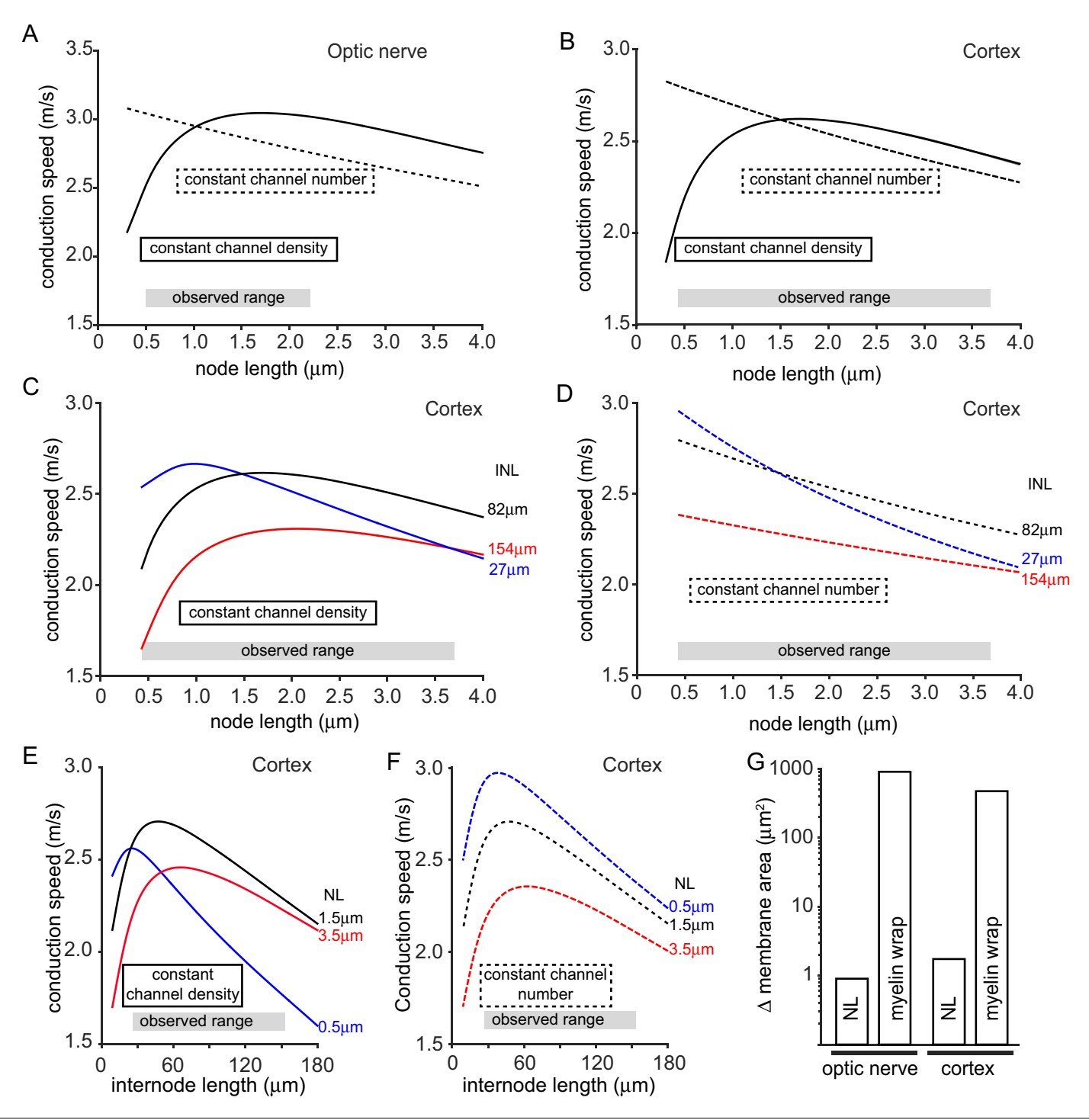

**Figure 3.** Predicted effect on conduction speed of different node lengths. (A–F) Calculated conduction speed as a function of node length for axons in (A) the optic nerve and (B–F) the cortical grey matter. For panels A–F, simulations were carried out assuming either that the density of ion channels at the node is constant as the node length is changed (solid lines), or that the number of ion channels is kept constant (dashed lines) at the value assumed for the mean node length. (C–D) Simulations for cortex as in B but examining the effect of altering internode length (INL, given by each curve). (E–F) Calculated dependence of conduction speed of cortical axons on internode length for different assumed node lengths (NL). The observed range of each abscissa parameter is indicated on the graphs. (G). Change in membrane area needed, in the myelin sheath (myelin wrap) or node of Ranvier (node length, NL), to change the conduction speed by 8.6% in optic nerve or 10.5% in cortical axons, when nodal ion channel density is held constant (note logarithmic scale).

speeds (<2% reduced) with variable length nodes, compared to when all the nodes along a particular axon had exactly the same length.

It has recently been reported that internode lengths may vary significantly in the cortex (*Tomassy et al., 2014*; *Chong et al., 2012*). We measured internode lengths in dye-filled axons in rat layer V cortex by measuring the distance between Na$_V$1.6 positive nodes. Similar to previous studies we found a large variability in the length of 30 internodes, the shortest being 27 μm and the largest 154 μm, with a mean value of 82.7 ± 6.3 μm. Given this large variation, we examined how internode length (assumed, for simplicity, to be the same for all internodes along an axon) affects the tuning of conduction speed by changes of node length (*Figure 3C–F*). For cortical axons (with a constant nodal sodium channel density), the peak of the dependence of conduction speed on node length is displaced to longer node lengths when the internode length is increased (*Figure 3C*), and in an axon with internode lengths of 154 μm the range of node lengths measured could generate changes in conduction speed of up to 42%. When the number of channels was held constant at each node, the dependence of conduction speed on node length was greater in axons with short internodes (27 μm), resulting in changes in conduction speed of up to 38% over the range of measured node lengths (*Figure 3D*). Changes of node length also affect the predicted dependence of conduction speed on internode length (*Figure 3E,F*). This relationship rises with internode length at low values of internode length, because more of the axon is myelinated, but decreases at large internode lengths as the spread of depolarization between nodes becomes less efficient. This relationship shows a sharper peak for shorter node lengths (*Figure 3E,F*).

We assessed how powerful node length changes can be for tuning conduction speed, compared to altering the amount of myelin around the axon. With the node length and internode length set at the mean values observed and the nodal conductances constant, if the number of wraps of myelin is decreased by 1 (from the normal 7 to 6 for the optic nerve and from the normal 5 to 4 for the cortex), the conduction speed is predicted to decrease by 8.6% for the optic nerve and 10.5% for the cortex (for simplicity no change of paranode length was assumed for these simulations, although the paranode may be shorter with less myelin wraps). For comparison, a similar speed decrease is predicted (with no change of wrap number) if the nodal Na$_V$ density is reduced by 26% in the optic nerve or 34% in the cortex, or if the internode length is increased by 36% in the optic nerve or 74% in the cortex. These same speed decreases can be achieved with no change of myelin wrapping or ion channel density by decreasing the node length from 1.02 μm to 0.625 μm in the optic nerve, or decreasing it from 1.5 μm to 0.635 μm in cortical axons. (Whether the internode needs to be lengthened by the same amount, to maintain axon length, and the consequences of this, are considered in the Simulations section of the Materials and methods). Strikingly, to produce these speed changes, the change of membrane area needed (shown in *Figure 3G*) when altering node length is 1006-fold less in the optic nerve, and 273-fold less in the cortex, than that needed when altering the myelin sheath. Thus, tuning conduction speed by altering node length is far more efficient than altering myelination when considering the amount of lipid and protein synthesis or breakdown needed, and would probably also be faster.

## Discussion

By combining anatomical measurements with computational modelling we have established a proof of principle for the idea that node of Ranvier length may be adjusted to tune axon conduction speed, and hence alter action potential arrival time. This differs from previous ideas on white matter plasticity (which have focussed on the effect of changes of myelin thickness and internode length (*Fields, 2008*; *Ullén, 2009*)), and demonstrates that the geometry of the node of Ranvier is also a crucial determinant of action potential conduction speed. Even at constant axon diameter, we found that node length displayed a surprising variability, both in the optic nerve and in the grey matter of the cortex (*Figure 1*). Remarkably, this node length variation was largely between different axons, while the nodes on any given axon tended to have similar lengths (*Figure 2*). The range of node lengths observed is sufficient to produce large variations in action potential conduction speed (*Figure 3*), comparable to those produced by adding or removing a wrap of myelin. These data suggest that axons may be able to adjust their node lengths in order to tune their conduction speed, and thus the arrival time of the information that they transmit.

There is now evidence of oligodendrogenesis and myelination in the adult CNS (*Dimou et al., 2008*; *Young et al., 2013*) which presumably results in the formation of new nodes of Ranvier. To avoid these, we studied only nodes that expressed the mature node marker Na$_V$1.6 (*Boiko et al., 2001*; *Kaplan et al., 2001*), and the fact that the variability in node length was observed across a large fraction of the nodes measured makes it improbable that this variation is solely due to nodes being at different developmental stages. The node length was approximately proportional to the amount of sodium channel labelling at the node (*Figure 1I*), suggesting that node length may mainly be adjusted by the insertion or removal of membrane containing sodium channels (although conceivably it is possible to vary node length in a manner independent of sodium channel trafficking by endo- or exocytosis of vesicles lacking sodium channels in their membrane).

The fact that node lengths are similar over long distances along an axon (*Figure 2B*) raises three mechanistic questions. First, when the node length is set to a different mean value in different axons, by what mechanism is the nodal ion channel density controlled (*Figure 1I*)? Second, what signal regulates node length, in order to adjust the arrival time of action potentials at the end of the axon? Conceivably a signal could be passed back along the axon from a postsynaptic cell by dynein-based motors, as occurs for BMP signalling from postsynaptic cells to the nuclei of presynaptic neurons (*Smith et al., 2012*). Third, what local molecular mechanism regulates the length of each node, how accurately can this be controlled, and is the internode length shortened when the node is elongated (to preserve overall axon length)? Node length regulation may involve modifying the paranodal cell adhesion between the myelin and the axon, mediated by the molecules Caspr, neurofascin 155 and contactin, as well as altering ankyrin G mediated scaffolding within the axon that locates voltage-gated Na$^+$ channels at the node (*Arancibia-Carcamo and Attwell, 2014*). Interestingly, nodal amyloid precursor protein has been proposed as a regulator of node length (*Xu et al., 2014*), prompting the speculation that changes in the processing of this molecule could alter node length in Alzheimer's disease.

Computer simulations of the propagation of action potentials along myelinated axons (*Figure 3*) show that rather small changes in node length can produce quite significant changes of conduction speed. The range of node lengths seen in the optic nerve (0.5–2.2 µm) can alter the conduction speed by up to 20% (for a constant nodal channel density in *Figure 3A*), while the range seen in the cortex (0.43–3.7 µm) can produce a larger change of up to 25% (for a constant density of nodal channels in *Figure 3B*) in axons with 82 µm long internodes, or 38% for axons with 27 µm long internodes (*Figure 3D*). The effect of altering node length is larger in cortical axons than in the optic nerve, partly because the 1.5-fold longer nodes contribute a larger fraction of the total axon capacitance in the cortex where the internodes are also 1.7-fold shorter.

Our data and simulations suggest that modulation of node length could be a viable strategy for adjusting the propagation time of action potentials to meet information processing needs. For an intercortical callosal axon of length 1 cm (in rat) or 6 cm (in human), with the properties that we assume for our simulations, a 20% decrease of axon conduction speed (from the value occurring for the mean observed node length) would increase the time needed to propagate information between the cortices from 3.8 to 4.8 ms in rat and from 23 to 29 ms in humans. Such modulation has been suggested to occur during chronic stress and major depression (*Miyata et al., 2016*) which shorten the node, as well as high frequency action potential activity (*Huff et al., 2011*; *Trigo and Smith, 2015*), acoustic over-exposure (*Tagoe et al., 2014*), pathological release of glutamate (*Fu et al., 2009*) and hypoxic conditions (*Reimer et al., 2011*), all of which lengthen the node. Altering node length offers the advantage that very small changes of membrane area, which could easily be produced rapidly by exocytosis or endocytosis at the node, produce large changes of conduction speed. In comparison, to produce the same speed changes by altering the number of myelin wraps requires the energetically expensive (*Harris and Attwell, 2012*), and probably more time consuming, synthesis or disassembly of a membrane area that is 273–1006 fold larger. In practice, both mechanisms might be used on different time scales.

## Materials and methods

### Electron microscopy

For the optic nerve, 3 male (8–10 weeks old) Sprague-Dawley rats were anaesthetised and perfused through the heart with fixative containing 2.5% glutaraldehyde and 2% paraformaldehyde in 0.1 M cacodylate buffer. The optic nerves were dissected out, post-fixed with 1% $OsO_4$ in 0.1 M cacodylate buffer, embedded in EPON and polymerised. Serial ultrathin (70 nm) sections, perpendicular to the nerve's long axis, were cut on an ultramicrotome and picked up on an osmium-coated glass slide. Back-scattered images were obtained on a scanning electron microscope (Hitachi SU8010) with a working distance of 2 mm, 1–1.5 kV accelerating voltage, scan speed of 40 or 80 s, and a typical pixel size of 1.65 nm at x30,000 magnification, and were analysed with ImageJ (FIJI). Node length (assessed from the number of sections containing the node) and mean axon diameter at the node (assessed as axon perimeter/$\pi$) were measured (uncorrected for tissue shrinkage during fixation).

### Tracer injections and node labelling

Four male 8–10 week old rats were anaesthetized with isoflurane and killed by cervical dislocation in accordance with United Kingdom animal experimentation regulations. After decapitation the brain was carefully dissected from the skull and 1 mm thick coronal slices containing the corpus callosum were obtained from the forebrain (from 4 to 8 mm rostral of the olfactory bulb) using a tissue cutter block. A 10% solution of tetramethylrhodamine dextran (MW 3000, Invitrogen, Paisley, UK) was iontophoretically injected into the cortical grey matter. Thereafter slices were incubated in oxygenated aCSF containing (in mM) 124 NaCl, 26 $NaHCO_3$, 1 $NaH_2PO_4$, 2.5 KCl, 2 $MgCl_2$, 2 $CaCl_2$, 10 glucose, bubbled with 95% $O_2$/5% $CO_2$ for 2 hr at room temperature to allow for diffusion of the tracer. After incubation slices were immersion fixed in PFA and resliced at 80–100 µm for subsequent immunohistochemical labelling of nodal ($Na_V1.6$) and paranodal (Caspr) marker proteins.

### Immunohistochemistry and confocal microscopy

Optic nerves and 4 mm thick coronal sections of fronto-parietal (motor) cortex (from 4 to 8 mm rostral of the olfactory bulb) from brains of 4 male (8–10 week old) Sprague-Dawley rats were either perfusion or immersion-fixed in 4% paraformaldehyde in PBS. Fixed tissue was then cut into 50 µm slices using a Leica vibratome VT1200S or, for $Na_V1.6$ density experiments, cut into 10 µm sections using a cryostat. Slices were blocked and permeabilised in 10% horse serum and 0.5% Triton X-100 in PBS. Immunofluorescence labelling was performed over 3 days with the following primary antibodies: rabbit anti-$Na_V1.6$ (Alomone, 1:500); mouse anti-Caspr clone K65/35 (Neuromab, UC Davies, 1:100). Slices were then washed extensively (3 $\times$ 20 min) and incubated overnight with secondary antibodies: anti-rabbit AlexaFluor488 (Invitrogen, 1:500), anti-mouse Dy-Light 647 (Jackson Immunoresearch, 1:500). Slices were then washed 3 $\times$ 10 min in PBS and mounted with Dako Fluorescent Mounting Medium. Slices were viewed using an LSM700 or LSM780 confocal microscope using a 63x (NA 1.4) oil immersion lens, and images were acquired with LSM software with the pinhole set to 1 Airy unit for the Caspr signal, resulting in an optical slice of 0.8 µm. Pixel size was 39.7 nm for *Figure 1A–H* and 99.2 nm for *Figure 1I* and 52.7 nm for node measurements in *Figure 2* and 263.6 nm for internode measurements in *Figure 2*.

### Image analysis

For node length analysis, confocal images were analysed using ImageJ software. Images were background subtracted and only nodes that lay approximately parallel to the plane of section (i.e. displayed nodal $Na_V1.6$ labelling with flanking Caspr-labelled paranodes all within a single 0.8 µm optical slice) were selected. Measuring the angle of the axon to the plane of the slice for a subset of 10 randomly chosen axons showed that the apparent node length measured in this way underestimated the actual node length by only 1.7% $\pm$ 0.6%. A maximum intensity projection was generated of the sections in which Caspr labelling was present for a particular node (up to five interleaved confocal slices at 0.38 µm intervals, with a maximum stack thickness of 2.32 µm), and a line intensity profile (the thickness of which was slightly less than the Caspr labelling thickness) was drawn spanning both Caspr-labelled paranodes. The size of the node was then calculated using a MATLAB (The MathWorks, Inc.) script which measures the distance between the half maximum intensity for each

paranode. Node diameter, paranode length and axon diameter were measured using the line tool in ImageJ over $Na_V1.6$ staining (for node diameter) and over the Caspr staining (for axon diameter and paranode length). $Na_V1.6$ staining was summed over the nodal area to obtain a parameter assumed to be proportional to sodium channel number. Internode length was measured in three dimensions in FIJI using the simple neurite tracer plugin (*Longair et al., 2011*). Data were not corrected for tissue shrinkage during fixation.

## Simulations

To simulate action potential propagation along myelinated axons, we implemented, in MATLAB, model C of *Richardson et al. (2000)* (at 37°C, with the unphysiologically low membrane capacitance of *Richardson et al. (2000)* corrected to a normal value). The differential equations of the model were derived and solved as in *Halter and Clark (1991)*. In brief, the axon is divided into compartments representing the node, paranode and internode. For each time step, current flow across the axonal or total myelin membrane is calculated from the values of voltage (and its rate of change), and the membrane capacitance and membrane conductances present per unit length (simultaneously solving the differential equations that define activation and inactivation of the voltage-gated currents present at the node), and intracellular and periaxonal axial current flow are calculated from the intracellular or periaxonal resistance per unit length and the gradient of intracellular or periaxonal voltage. Details of the parameters used are summarised in *Table 1*. The MATLAB code used can be obtained immediately on request from the authors; it will be written up and documented as a resource for free access from GitHub by August 1st 2017.

Simulations were carried out as in *Bakiri et al. (2011)* except that the periaxonal space under the myelin was included (51 nodes were simulated and conduction speed was measured between nodes 20 and 30). This model includes fast and persistent $Na^+$, and slow $K^+$, voltage-gated channels at the node, but omits voltage-gated $K^+$ channels at the juxtaparanode (which are little activated because the 100 mV voltage change of the action potential is distributed across the 11–15 membranes of the 5–7 myelin wraps and the axon, implying only a 7–9 mV voltage change across the axonal membrane). For simplicity, the node length was usually assumed to be the same at all nodes on the axon, i.e. we ignored the variability in node length along the same axon described in *Figure 2*. The node diameter was set to the mean value measured experimentally, i.e. 0.73 µm (in 164 nodes) for the optic nerve, and 0.64 µm (in 158 nodes) for cortex. The region between two nodes, 139.3 µm long for the optic nerve (*Butt et al., 1994*) and 81.7 µm long for the cortical axons (the mean value measured in layer V from 30 internodes, see main text), was kept constant when node length was varied (both for simplicity, and because there was no correlation of node length and internode length: *Figure 2E*). This internodal region was divided (along its length) into 66 and 86 compartments for the optic nerve and the cortex, respectively, the end 2.11 and 1.90 µm parts of which represent the paranode where the myelin attaches to the axon (values measured in 164 and 158 nodes, respectively, from the length of the Caspr labelling; the number of compartments used has to be large for the simulation to be accurate, and needs to be chosen so that an integral number of compartments can represent the paranodal junction; the number was adjusted appropriately when simulating different internode lengths). The internodal axon diameter is larger than the diameter at the node (*Halter and Clark, 1991*; *Berthold and Rydmark, 1983*), although this difference is a much smaller percentage for small than for large axons (*Rydmark and Berthold, 1983*). The internodal and paranodal axon diameters were set to 0.82 µm and 0.73 µm for the optic nerve and cortex, respectively (mean values obtained from 164 axons in optic nerve and 158 cortical axons from the diameter of the paranodal Caspr labelling), so that the node diameters were 88% and 86% of the internodal axon diameters respectively. Apart from at the paranodes, the internodal axon was assumed to be surrounded by a periaxonal space of thickness 15 nm (*Robertson, 1959*; *Mierzwa et al., 2010*; *Möbius et al., 2016*), and to have a g ratio (axon diameter/myelin diameter) of 0.79 in the optic nerve and 0.8 in the cortex (*Sugimoto et al., 1984*; *Oorschot et al., 2013*; to obtain an integral number of myelin wraps these values were adjusted slightly, to 0.78 and 0.81 respectively). This led to the optic nerve and cortical axons having 7 and 5 myelin wraps, respectively assuming a myelin wrap periodicity of 15.6 nm (*Agrawal et al., 2009*; *Harris and Attwell, 2012*). The periaxonal space at the paranode, because of the structure of the attachment of the myelin to the axon at the paranode, is thought to comprise (*Mierzwa et al., 2010*) a pathway of cross sectional area A = 170 $nm^2$, which spirals around the axon (from the node to the periaxonal space of the internode) for a total

distance of approximately D = π.d.$N_{wraps}$ where $N_{wraps}$ is the number of myelin wraps and d is the axon diameter. A periaxonal space of width w, along a paranode of length L, would have the same resistance as this pathway if

$$L/(\pi.d.w) = D/A \text{ or } w = A.L/(\pi.D.d) = A.L./[(\pi.d)^2.N_{wraps}]$$

The effective value of w used to model this spiral pathway for the optic nerve and cortical axons was thus 0.0077 nm and 0.0123 nm respectively.

In the main text we present calculations showing that a given change of conduction speed can be produced far more efficiently (in terms of the change of membrane area needed) by shortening of the node than by adding another wrap of myelin. Those calculations ignore the possibility that, when the node is shortened, the internode needs to be lengthened by the same amount in order to maintain the axon length. It is unclear whether the sub-micron node length changes postulated in our calculation would actually require remodelling of the adjacent myelin sheath – conceivably slackness in the somewhat non-straight internode would allow the change of node length to be accommodated without a change of internode length, and furthermore node length might change by an eversion of the paranodal loops closest to the node without any other significant change to the myelin sheath (reviewed by *Arancibia-Carcamo and Attwell, 2014*). Thus, small changes in nodal length might well occur without major remodelling of the myelin sheath. Nevertheless, if one assumes that node shortening by X µm absolutely does require an X µm elongation of the myelin sheath, then more membrane changes are needed than are accounted for in our simple calculation. One can show mathematically that the sheath membrane area increase is larger than the area decrease at the node by a factor of 2x(number of myelin wraps)x(mean radius of wraps)/(node radius), which is roughly 18.4 for the optic nerve and 13 for the cortex. Accounting for these area changes (and noting that, in this situation, there is no change of total axon length) would reduce the ratio of the membrane area changes needed to produce a given speed change (when adding a layer of myelin to the sheath versus changing the node length) from 1006-fold to 55-fold for the optic nerve and from 273-fold to 21-fold for the cortex, but these ratios remain impressively large, and so the energetic argument favouring speed tuning by alteration of the node length still holds.

## Statistics

Data are shown as mean±s.e.m. Comparisons are via 2-tailed Student's t-tests unless otherwise stated. Assessment of whether the slope of linear regressions differed significantly from zero was obtained using the t-statistic for the slope.

## Acknowledgements

Supported by a Wellcome Trust Senior Investigator Award to DA, a Wellcome Trust PhD studentship to LC, a Marie Curie fellowship to MCF, and JSPS grants (24650181 and 25245069) to KT. We thank Boris Barbour, Beverley Clark, Renaud Jolivet, Josef Kittler, Anna Krasnow and Angus Silver for comments on the manuscript.

## Additional information

### Funding

| Funder | Grant reference number | Author |
| --- | --- | --- |
| Wellcome Trust | 099222/Z/12/Z | I Lorena Arancibia-Cárcamo<br>Lee Cossell<br>David Attwell |
| European Commission | 623714 AxonGliaPlasticity | Marc C Ford |
| Japan Society for the Promotion of Science London | 24650181 | Kinji Ishida<br>Koujiro Tohyama |
| Japan Society for the Promotion of Science London | 25245069 | Kinji Ishida<br>Koujiro Tohyama |

The funders had no role in study design, data collection and interpretation, or the decision to submit the work for publication.

### Author contributions

ILA-C, Conceptualization, Data curation, Software, Formal analysis, Funding acquisition, Validation, Investigation, Visualization, Methodology, Writing—original draft, Writing—review and editing; MCF, Conceptualization, Data curation, Formal analysis, Funding acquisition, Validation, Investigation, Visualization, Methodology, Writing—original draft, Writing—review and editing; LC, Conceptualization, Resources, Data curation, Software, Formal analysis, Validation, Investigation, Visualization, Methodology, Writing—original draft, Writing—review and editing; KI, Investigation, Visualization, Methodology; KT, DA, Conceptualization, Resources, Data curation, Software, Formal analysis, Supervision, Funding acquisition, Validation, Investigation, Visualization, Methodology, Writing—original draft, Project administration, Writing—review and editing

### Author ORCIDs

Marc C Ford, http://orcid.org/0000-0003-0472-2652
David Attwell, http://orcid.org/0000-0003-3618-0843

### Ethics

Animal experimentation: This work was performed in accordance with the United Kingdom Animals (Scientific Procedures) Act (1986) and subsequent amendments, under UK Home Office licence (PPL 70/7299) and the electron microscopy part of the work was approved by the Ethical Committee of Iwate Medical University.

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
