## [Decision Letter]

[Editors’ note: a previous version of this study was rejected after peer review, but the authors submitted for reconsideration. The first decision letter after peer review is shown below.]

Thank you for choosing to send your work entitled "Ranvier node length and myelinated axon conduction speed" for consideration at *eLife*. Your full submission has been evaluated by Eve Marder (Senior Editor), a Reviewing Editor, and two peer reviewers, and the decision was reached after discussions between the reviewers. Based on our discussions and the individual reviews below, we regret to inform you that we are forced to reject this version of the manuscript.

The reviewers were quite conflicted because they really were taken by the aims and goals of the paper, and thought it is potentially important. Nonetheless, they were troubled by a variety of technical issues that are perhaps captured by their feeling the manuscript suffered by comparing "apples and oranges". Virtually all of the reviewers' comments could be dealt with, but *eLife* has a strong policy to not require extensive new experimental work when allowing a revision. Therefore, when reviewers really ask for extensive experimental work (work that we expect would take more than a month to do), we are forced to reject the manuscript. This frees the authors to take their manuscript elsewhere if they choose, as it is no longer in consideration at *eLife*. If, however, you feel in retrospect that the reviewers are, on balance, correct, and if you can deal with these critiques, we would be willing to entertain a new submission, which would be treated as such, at some time in the future.

*Reviewer #1:*

This brief manuscript combines an assessment of the variation in length of nodes of Ranvier (the distance separating adjacent internodes) along myelinated axons in the rat CNS with modeling to estimate what effect these variations might have on the conduction velocity of action potentials. The extent and character of myelin is regulated by neuronal activity during development, and there has is renewed interest in the possibility that myelination may be modified throughout life. The effects of altered myelination could be exerted at many levels – myelinated versus unmyelinated, thickness of myelin, length of internodes, and length of nodes, not to mention changes in NaV and Kv densities along axons. However, little is known about the variations in these parameters in the adult CNS. Defining these variations, exploring how they are affected by life experience and determining their effects on information processing are important goals. Here the authors show that there is greater variation in node length along myelinated axons in brain than in optic nerve, and through modeling show that this variation in brain would be expected to have substantial effects on conduction velocity.

1) The study uses two different methods to determine the length of nodes – EM data from optic nerve and immunofluorescence for brain. It isn't clear why different methods were used for the two preparations. Some assurance should be provided that the differences seen are not due to the use of different methodologies. In particular, the authors should estimate what error might be induced by analyzing nodes that are in different orientations in the brain samples and indicate how orientation was determined. It would be helpful if a dozen examples spanning the range of node lengths were illustrated in Figure 1.

2) The study compares tissue from animals that were different ages – 8-12 weeks for the optic nerve and P30 for the brain. It isn't clear why different ages were examined. Oligodendrogenesis and myelination are still ongoing at P30 in brain, raising the possibility that the variation in node length reflects an intermediate developmental state. As internodes grow together to form nodes, some of the longer nodes could represent internodes captured at this stage. As the development of myelin varies across cortical layers and in different regions of the cortex, the authors should provide information about the regions of the cortex that were examined.

3) As NaV density is a crucial parameter, it would be valuable to examine NaV immunoreactivity at these nodes. Although not quantitative, this could provide some indication of whether NaV density or NaV number is held constant. Also, NaV staining would help clarify if the longest nodes are merely two hemi-nodes prior to consolidation.

4) If node lengths varied from 0.3 to 4 μm in the cortex, it appears that only a partial dataset is shown in Figure 1 or that the x-axis is mis-labeled.

5) The plot on Figure 3 should extend from.4 -1.4, because this is the range of node lengths observed, or at least, this region should be highlighted in the graph.

6) It seems most relevant to highlight the effect that the observed changes in node length would have on conduction velocity in Figure 3. In particular, although the range is larger in brain, the distribution in Figure 2 suggests that ~90% of nodes fall within 0.5 to 2 um, which the modeling suggests would have minimal effect on conduction velocity.

7) The prediction about the length of successive nodes on the same axon should be validated by examining node length along individual axons using available serial EM (TEM or SEM) datasets.

*Reviewer #2:*

Arancibia-Carcamo et al., submit "Ranvier node length and myelinated axon conduction speed" for consideration at *eLife*. Numerous recent studies have indicated that myelin in the central nervous system can be generated and modulated well in to adult life. The manuscript by Arancibia-Carcamo et al., serves as a timely reminder that modulation of myelin is not the only way in which the functional output of myelinated axons can be modulated and fine-tuned. The study is primarily a modeling based study built on the observations that nodes of Ranvier vary in length (independently of axon/ node diameter) in both the optic nerve, and to a greater extent, the cerebral cortex. The main predictions of the model are that relatively subtle changes in node of Ranvier length can profoundly affect conduction speed, and that such nodal changes are likely very energy efficient compared to the changes in myelin thickness that would be required to elicit similar effects on speed. This core conclusion/prediction is certainly provocative and interesting, and provides a simple hypothesis that will require experimental investigation in future experiments out-with the scope of this manuscript. However, the manuscript itself needs significantly more detail in parts before it is ready for publication, and could some additional considerations could be integrated into the model to provide more depth. The study is, in my opinion, of sufficient general interest to the readership of *eLife*.

1) Ultimately there are two main concepts here, that changing nodal length can affect conduction speed and that this may be a far more energy efficient way to fine tune function than modulating myelination. The first concept has been delineated nicely given the parameters applied, but the second dealt with a little more cursorily (by simply pointing out that different membrane areas would be involved in remodelling a node versus myelin). Could the authors consider the following points in their model?

A) What would the effect of conduction be on simply changing ion channel density at the node and keeping node length the same? One could imagine that the targeting of channels to nodes of a similar size might represent an even more straightforward manipulation that required no change in membrane area at all. How could this effect conduction speed?

B) Could the authors take into account the consequences of changing node length along the length of the axon and take these into consideration with respect to energy requirements (in which the corresponding author is also expert)? If, for example, an average myelin sheath length in the cortex is say 29um+ 1um for the node and the axon is 3mm long then there would be 100 myelin sheaths along its length. If a 1um node were to increase in length to 3um then 200um of space along the axon would be required to accommodate the longer nodes. This would still require significant remodelling of myelin along the axon, either through small changes in many sheaths or the removal of a small number of sheaths and subsequent readjustment.

I don't know if the authors can easily integrate such considerations into their model and account for relevant energy requirements, but the point is that the simple comparison of membrane area seems quite simplistic, particularly given that this is a core conceptual point of the study.

2) The diameter of the axon is constricted at the node and in particular at the paranodal axon-glial junction. The authors have not really dealt with this point. They mention that values as to the relative axonal sizes have been determined by "unpublished observations of 10-12 axons," but it is not clear how many nodes this reflects, nor more importantly what the variability in internode to node diameter is. Perhaps this is a very important parameter that could profoundly affect functional output and is it being overlooked here. More details and consideration will be important, particularly as at least one previous study has investigated how constriction of diameter at the node affects conduction.

3) There are essentially no details relating to the model itself in the manuscript. Input values for various parameters are provided and one sentence referring to modifications of previous models described in two references. Without getting in to the details of the Richardson et al. paper in particular readers of *eLife* will be left quite in the dark about the details of the model being used and indeed of the alternatives. The pros and cons of different models have not been discussed, and so it is impossible to review the modelling aspect per se. The authors should provide the details of each model, at least as an appendix or as supplementary data, giving due recognition to other sources as required?

This is clearly not essential, but, if appropriate, would it be possible for the authors to generate an online interface so that the community could take advantage of the model? This could be an incredible resource.

[Editors’ note: what now follows is the decision letter after the authors submitted for further consideration.]

Thank you for resubmitting your work entitled "Node of Ranvier length and myelinated axon conduction speed" for further consideration at *eLife*. Your article has been favorably evaluated by Eve Marder (Senior Editor) and three reviewers, one of whom is a member of our Board of Reviewing Editors.

The manuscript has been improved but there are some remaining issues that need to be addressed before acceptance, as outlined below:

The essential experimental findings of this resubmitted manuscript are that in CNS nodal length varies over an approximate 4.4 to 8.7-fold range, but is more consistent along individual axons. Little correlation is seen between nodal length and width, which is taken as evidence that nodal length can be regulated at an axon-specific level independently of axonal diameter. Computer modeling predicts that the range of nodal lengths observed will have a biologically significant effect on conduction (in the order of ~20-38%). The authors therefore propose that regulation of nodal length is a viable method for fine-tuning conduction velocity. They make a compelling case for it being a less energy/resource demanding form of plasticity than others currently under investigation (e.g., myelin thickness, de novo myelination). The paper is thought-provoking and of sufficient interest for the readership of e*Life*.

In general the manuscript is considerably improved, being more detailed and streamlined. The authors have addressed the initial comments in detail. Much of the in vivo relevance relies on the strength of the new data where individual axons have been labelled and along which node length has been measured to determine relative variability along and between axons. This is a very nice approach and the authors should be commended for this revision. Although many of the concepts proposed will require extensive experimental investigation, this study in and of itself is important in that it provides numerous key hypotheses that can and will be tested in the future.

1) A new reviewer (on the second round) notes: I am a little reluctant to shift the goalposts on the authors who seem to have responded fairly well to the first set of reviewer comments. Nevertheless, I feel that the language of the manuscript sometimes goes beyond what is demonstrated (e.g., "Thus, axon-specific adjustment of node of Ranvier length is an energy-efficient and potentially rapid mechanism for tuning the arrival time of information in the CNS"). It is not shown that nodal length is modified during plasticity, or even that it varies between relevant axons in naturally occurring systems where adaptations need to be made to ensure synchrony (such as in auditory brainstem circuits, where the authors have previously studied internode length and axonal diameter). In the absence of such data, it should be made very clear in the Abstract that this paper identifies regulation of nodal length as a potential mechanism for neuroplasticity only. Indeed, the authors must make sure that also the superficial reader will not confuse fact and fiction. "Fact" and well documented is the observation that nodes are more similar in size along an individual axon than the nodes of different but otherwise comparable axons. Still "fiction" is the idea that nodes can change in size (with predictable consequences on axonal conduction velocity) as a physiological mechanism, by which neurons adapt axonal conduction to the demands of a larger neuronal network. For comparison, had the authors been the first to discover that axons differ in caliber – which of course is well known – we could have the same discussion. In fact, it might be easier for neurons to fine-tune their axonal calibers (e.g. by Akt/mTOR signaling) than to remodel nodal domains individually in a concerted effort with oligodendrocytes. Thus, there was consensus of all reviewers that the language needs to make much clearer (also in title and Abstract) that the main conclusions that make the paper so interesting are hypothetical/ theoretical.

2) Figure 2: One wonders how the relative variability along versus between axons would look without inclusion of the two axons with longer nodal lengths ("outliers", late developmental stages? See below). One worries to what extent the increased variance between axons is driven by these two axons. Perhaps the authors can simply address this comment without further information, but the presentation and annotation/ explanation of Figure 2 is difficult to interpret. If the removal of the two axons with the larger nodes removes the differences between versus within axons then it would be reassuring to see a slightly larger number of axons such that one didn't worry about any outlier or sampling artifacts. How representative are the 3 axons shown in Figure 2? Can the authors address this issue by analysis of additional axons (in the hope of catching more that have nodal lengths in the 2-3μm range) or by a separate analysis that excludes these two axons)?

3) In response to the concern that nodes may be still be immature at P30, the authors repeated their analysis with cortices at age 8-10 weeks. However, cortical myelination may not be finished by 2 months either, because even at age 6 months (NG2CreERT2-based) OPC lineage tracing experiments revealed newly generated oligodendrocytes (Dimou et al., J Neurosci. 2008) that presumably make myelin with immature nodes of Ranvier. This should be discussed.

---

## [Author Response]

[Editors’ note: the author responses to the first round of peer review follow.]

[…] Reviewer #1:

*[…] 1) The study uses two different methods to determine the length of nodes – EM data from optic nerve and immunofluorescence for brain. It isn't clear why different methods were used for the two preparations. Some assurance should be provided that the differences seen are not due to the use of different methodologies.*

We have now carried out an analysis of the length of nodes of Ranvier in the optic nerve using immunofluorescence to compare to the existing EM data. We find that the mean node length is similar using fluorescence and EM, but the total range of lengths observed is twice as large in fluorescently labeled tissue as is seen in EM. This probably reflects the larger sample size studied with immunofluorescence. This has been discussed in the manuscript (subsection “Variation of node length in the optic nerve and cortex”, second paragraph).

*In particular, the authors should estimate what error might be induced by analyzing nodes that are in different orientations in the brain samples and indicate how orientation was determined.*

In order to minimize the error induced by analyzing nodes that are in different orientations in the sections, only nodes that lay parallel to the plane of optical sections were measured. This has now been made clear in the Materials and methods section (subsection “Image Analysis”).

*It would be helpful if a dozen examples spanning the range of node lengths were illustrated in Figure 1.*

We thank the reviewer for this helpful suggestion. We have completely redesigned Figure 1 to include nodes of various lengths for both the optic nerve and the cortex, and an analysis of node length variation along single axons is presented in Figure 2.

2) The study compares tissue from animals that were different ages – 8-12 weeks for the optic nerve and P30 for the brain. It isn't clear why different ages were examined. Oligodendrogenesis and myelination are still ongoing at P30 in brain, raising the possibility that the variation in node length reflects an intermediate developmental state. As internodes grow together to form nodes, some of the longer nodes could represent internodes captured at this stage. As the development of myelin varies across cortical layers and in different regions of the cortex.

We originally used different ages because those were the ages of suitably labelled tissue that were readily available to us. In order to show that the variation in node length is not due to imaging nodes at various stages of development, but due to an actual diversity in the length of mature nodes, we have now re-done all the immunocytochemistry in 8-10 week old rats (to match the age group used for the EM analysis of optic nerve). Furthermore, we confirmed that all the nodes of Ranvier that were imaged and analysed were positive for the mature node marker NaV1.6. These methodological points are stated in the main text (NaV1.6 labelling: subsection “Variation of node length in the optic nerve and cortex”, second paragraph) and in the Materials and methods (animal age: subsections “Tracer injections and node labelling” and “Immunocytochemistry and confocal microscopy”).

*The authors should provide information about the regions of the cortex that were examined.*

We apologise that this was not made clear in the previous version of the manuscript. All cortical imaging was done in layer 5 of the motor cortex of frontoparietal cortex. This has now been specified in the main text (subsection “Variation of node length in the optic nerve and cortex”, third paragraph) and in the Materials and methods section (subsection “Immunocytochemistry and confocal microscopy”).

*3) As NaV density is a crucial parameter, it would be valuable to examine NaV immunoreactivity at these nodes. Although not quantitative, this could provide some indication of whether NaV density or NaV number is held constant. Also, NaV staining would help clarify if the longest nodes are merely two hemi-nodes prior to consolidation.*

As indicated above we have now only included in our analysis nodes which were positive for the mature node marker NaV1.6, in order to eliminate the possibility of them being immature nodes. In addition, as suggested, we have now carried out an analysis of NaV1.6 immunoreactivity (summed over the node area) as a function of node length. Overall there is a strong positive correlation between node length and total NaV labelling, suggesting that NaV density is maintained constant in nodes of different length. However, at any specific node length there is a large variability in NaV1.6 staining, suggesting NaV density is not very tightly regulated, and perhaps there are situations in which nodes may change size without altering their number of NaV channels. This has been discussed in the Results (subsection “Variation of node length in the optic nerve and cortex”, last paragraph; subsection “Predicted effects of node length variation on conduction speed”, second paragraph) and Discussion of the manuscript (first paragraph), and simulations have been provided to cover both situations.

*4) If node lengths varied from 0.3 to 4 μm in the cortex, it appears that only a partial dataset is shown in Figure 1 or that the x-axis is mis-labeled.*

We apologise for confusing the referee. In the original Figure 1 the node lengths were displayed on the y axis (rather than the x axis as was employed in the original Figure 2 and Figure 3), and they did indeed cover the full observed range from 0.3 to 4 µm. However, in the light of our new data we have now completely redesigned Figure 1, and in the plots displaying node lengths in optic nerve and cortex layer V (Figure 1) node length is now plotted on the x axis.

*5) The plot on Figure 3 should extend from.4 -1.4, because this is the range of node lengths observed, or at least, this region should be highlighted in the graph.*

We thank the reviewer for this suggestion. We now indicate the range of observed node lengths (or internode lengths) in all the graphs displaying the results of the simulations (Figure 3).

*6) It seems most relevant to highlight the effect that the observed changes in node length would have on conduction velocity in Figure 3. In particular, although the range is larger in brain, the distribution in Figure 2 suggests that ~90% of nodes fall within 0.5 to 2 um, which the modeling suggests would have minimal effect on conduction velocity.*

Figure 3 has now changed significantly compared to our previous submission. In the first submission we assumed an internode length of 31 µm which was measured from a small sample of internodes assessed by MBP staining. Using tracer injections into brain slices has now allowed us to measure a greater number of internodes and we now find that the mean internode length in the cortex is closer to 82 µm. This is similar to that observed in the literature (e.g. Chong et al., 2011, Figure 1). Furthermore, as observed by Tomassy et al. (2014) and Chong et al. (2011), we find great variability in internode length (27-154 µm range in layer V of rat cerebral cortex). The new Figure 3 shows how the effect of node length variation changes depending on internode length. Importantly, with our new parameters we find that the greatest effect on conduction velocity occurs in nodes with lengths between 0.5-2 µm, which (as the reviewer correctly highlights) is where the vast majority of nodes fall (Figure 3).

*7) The prediction about the length of successive nodes on the same axon should be validated by examining node length along individual axons using available serial EM (TEM or SEM) datasets.*

We appreciate the suggestion of validating the prediction about the length of successive nodes. Available serial EM datasets, however, do not cover a range long enough for us to be able to follow axons for the length of several long internodes. In order to overcome this difficulty, we have now added new experiments which combine single axon tracing (using a fluorescent dextran tracer) with immunohistochemical labelling of nodal (NaV1.6) and paranodal (Caspr) marker proteins. This approach enabled us to analyse the length of up to 13 consecutive nodes of Ranvier along individual axons over 1 mm long (see new Figure 2). Remarkably, our new data demonstrate that node length is similar along individual axons but different between axons, which strongly suggests that individual axons independently adjust their node lengths to tune conduction speed.

Reviewer #2:

*[…] 1) Ultimately there are two main concepts here, that changing nodal length can affect conduction speed and that this may be a far more energy efficient way to fine tune function than modulating myelination. The first concept has been delineated nicely given the parameters applied, but the second dealt with a little more cursorily (by simply pointing out that different membrane areas would be involved in remodelling a node versus myelin). Could the authors consider the following points in their model?*

*A) What would the effect of conduction be on simply changing ion channel density at the node and keeping node length the same? One could imagine that the targeting of channels to nodes of a similar size might represent an even more straightforward manipulation that required no change in membrane area at all. How could this effect conduction speed?*

We thank the reviewer for this suggestion. Changing the number of ion channels on the membrane might be difficult without altering the amount of membrane at the node, as their insertion would be dependent on membrane trafficking. It is because of this that we have simulated two scenarios: one in which the node elongates as more NaV channels are inserted (i.e. keeping the same density of channels) and one in which only membrane is inserted (keeping the same channel number). Experimentally:

i) there is a large variation in node length (Figure 1 and Figure 2), so adjusting the channel number without changing the membrane area does not seem to be a strategy employed by the cells, and

ii) our new experiments added to Figure 1 suggest that the density is kept approximately constant as more membrane is added to lengthen the node of Ranvier.

However, to provide a comparison for the reader, we have now added an example in the last paragraph of the Results section: “For comparison, a similar speed decrease is predicted (with no change of wrap number) if the nodal Na_V_ density is reduced by 26% in the optic nerve or 34% in the cortex…”.

*B) Could the authors take into account the consequences of changing node length along the length of the axon and take these into consideration with respect to energy requirements (in which the corresponding author is also expert)? If, for example, an average myelin sheath length in the cortex is say 29um+ 1um for the node and the axon is 3mm long then there would be 100 myelin sheaths along its length. If a 1um node were to increase in length to 3um then 200um of space along the axon would be required to accommodate the longer nodes. This would still require significant remodelling of myelin along the axon, either through small changes in many sheaths or the removal of a small number of sheaths and subsequent readjustment.*

For our simulations, internode length was maintained constant in order to isolate clearly for the reader the effect of node length on conduction velocity. It is unclear whether the sub-micron node length changes postulated in our calculation (of the membrane areas changing) would actually require remodelling of the adjacent myelin sheath – conceivably, slackness in the somewhat non-straight internode would allow the change of node length to be accommodated without a change of internode length, and furthermore node length might change by an eversion of the paranodal loops closest to the node without any other significant change to the myelin sheath (reviewed by Arancibia-Carcamo and Attwell, 2014, see Figure 3 and associated text). Thus, small changes in nodal length might well occur without major remodelling of the myelin sheath.

Nevertheless, if one assumes that node shortening by X µm absolutely does require an X µm elongation of the myelin sheath, then it is true that more membrane changes are needed than are accounted for in our simple calculation. One can show mathematically that the sheath membrane area increase is larger than the area decrease at the node by a factor of 2x(number of myelin wraps)x(mean radius of wraps)/(node radius), which is roughly 18.4 for the optic nerve and 13 for the cortex. Accounting for these area changes (and noting that, in this situation, there is no change of total axon length) would reduce the ratio of the membrane area changes needed to produce a given speed change (when adding a layer of myelin to the sheath versus changing the node length) from 1006-fold to 55-fold for the optic nerve and from 273-fold to 21-fold for the cortex, but these ratios remain impressively large, and so the energetic argument favouring speed tuning by alteration of the node length still holds. We have now discussed all this in the last paragraph of the subsection “Simulations”, and flagged that discussion in the first paragraph of the Discussion section.

*2) The diameter of the axon is constricted at the node and in particular at the paranodal axon-glial junction. The authors have not really dealt with this point. They mention that values as to the relative axonal sizes have been determined by "unpublished observations of 10-12 axons," but it is not clear how many nodes this reflects, nor more importantly what the variability in internode to node diameter is. Perhaps this is a very important parameter that could profoundly affect functional output and is it being overlooked here. More details and consideration will be important, particularly as at least one previous study has investigated how constriction of diameter at the node affects conduction.*

We apologise if this was not clear enough: the diameter of the axon is indeed constricted at the node and this has always been taken into consideration in our simulations (as described in the subsection “Simulations”). In our revised manuscript we have now collected new data, and have measured the diameter of the axon at the paranodes as well as the nodes for all the nodes measured (158 nodes for the cortex and 164 nodes for the optic nerve). These parameters (given in Table 1) were included in the model when carrying out the new simulations presented in this manuscript. There is of course some variability in the ratio of the internode diameter to node diameter, but the focus of this paper is on the effect of node length, and diversifying the discussion to consider every parameter of the axon will detract from our main message.

*3) There are essentially no details relating to the model itself in the manuscript. Input values for various parameters are provided and one sentence referring to modifications of previous models described in two references. Without getting in to the details of the Richardson et al. paper in particular readers of eLife will be left quite in the dark about the details of the model being used and indeed of the alternatives. The pros and cons of different models have not been discussed, and so it is impossible to review the modelling aspect per se. The authors should provide the details of each model, at least as an appendix or as supplementary data, giving due recognition to other sources as required?*

The model is now described in more detail in the Materials and methods subsection “Simulations”.

*This is clearly not essential, but, if appropriate, would it be possible for the authors to generate an online interface so that the community could take advantage of the model? This could be an incredible resource.*

As stated in the new Methods section (subsection “Simulations”, second paragraph), the MATLAB code used can be obtained on request from the authors; we are in the process of writing it up and documenting it as a resource for free access by others, and it will be deposited on ModelDB and Github in the next 6 months.

[Editors' note: the author responses to the re-review follow.]

*[…] 1) A new reviewer (on the second round) notes: I am a little reluctant to shift the goalposts on the authors who seem to have responded fairly well to the first set of reviewer comments. Nevertheless, I feel that the language of the manuscript sometimes goes beyond what is demonstrated (e.g., "Thus, axon-specific adjustment of node of Ranvier length is an energy-efficient and potentially rapid mechanism for tuning the arrival time of information in the CNS"). It is not shown that nodal length is modified during plasticity, or even that it varies between relevant axons in naturally occurring systems where adaptations need to be made to ensure synchrony (such as in auditory brainstem circuits, where the authors have previously studied internode length and axonal diameter). In the absence of such data, it should be made very clear in the Abstract that this paper identifies regulation of nodal length as a potential mechanism for neuroplasticity only. Indeed, the authors must make sure that also the superficial reader will not confuse fact and fiction. "Fact" and well documented is the observation that nodes are more similar in size along an individual axon than the nodes of different but otherwise comparable axons. Still "fiction" is the idea that nodes can change in size (with predictable consequences on axonal conduction velocity) as a physiological mechanism, by which neurons adapt axonal conduction to the demands of a larger neuronal network. For comparison, had the authors been the first to discover that axons differ in caliber – which of course is well known – we could have the same discussion. In fact, it might be easier for neurons to fine-tune their axonal calibers (e.g. by Akt/mTOR signaling) than to remodel nodal domains individually in a concerted effort with oligodendrocytes. Thus, there was consensus of all reviewers that the language needs to make much clearer (also in title and Abstract) that the main conclusions that make the paper so interesting are hypothetical/ theoretical.*

The language has now been changed throughout to unequivocally distinguish our experimental conclusions and our theoretical conclusions. In addition, following the reviewers’ suggestion, although we thought the original title (“Node of Ranvier length and myelinated axon conduction speed”) was completely neutral, we have now emphasised the fact that we have not yet demonstrated changes of node of Ranvier length during learning by changing the title of the manuscript to “Node of Ranvier length as a potential regulator of myelinated axon conduction speed”, with a similar change being made in the Abstract.

*2) Figure 2: One wonders how the relative variability along versus between axons would look without inclusion of the two axons with longer nodal lengths ("outliers", late developmental stages? See below). One worries to what extent the increased variance between axons is driven by these two axons. Perhaps the authors can simply address this comment without further information, but the presentation and annotation/ explanation of Figure 2 is difficult to interpret. If the removal of the two axons with the larger nodes removes the differences between versus within axons then it would be reassuring to see a slightly larger number of axons such that one didn't worry about any outlier or sampling artifacts. How representative are the 3 axons shown in Figure 2? Can the authors address this issue by analysis of additional axons (in the hope of catching more that have nodal lengths in the 2-3μm range) or by a separate analysis that excludes these two axons)?*

For Figure 2 we traced 18 axons and measured the node length of as many nodes as we could find along those axons. The 3 axons shown were chosen to illustrate different mean node lengths. As the reviewer points out, the majority of axons had nodes that fell within the 0.5-2 µm range, which is consistent with the distribution of node lengths shown in Figure 1. We found 2 axons in which the nodes were of longer lengths (2-3.5 µm), which is at the frequency expected from the probability distribution in Figure 1 where ~10% of nodes fall into this bracket. Thus, these 2 axons are not necessarily “outliers” or developing nodes (see below) but just what would be expected from the variation in node length across axons reported here. Nonetheless, following the reviewer’s suggestion, we have calculated the coefficient of variation excluding these 2 axons. We find that, even with these two axons excluded, the mean coefficient of variation for the node length along single axons is 38.8 ± 4.7% lower than that across all axons (highly significantly different, with a p value of 6.6x10^-7^) which strongly reinforces our original conclusion that node length variability is far greater across axons than within axons. This has now been reported in the Results section (subsection “Node lengths vary far more between axons than along axons”, first paragraph).

*3) In response to the concern that nodes may be still be immature at P30, the authors repeated their analysis with cortices at age 8-10 weeks. However, cortical myelination may not be finished by 2 months either, because even at age 6 months (NG2CreERT2-based) OPC lineage tracing experiments revealed newly generated oligodendrocytes (Dimou et al., J Neurosci. 2008) that presumably make myelin with immature nodes of Ranvier. This should be discussed*.

We agree that cortical, or even optic nerve (Young et al., 2013), myelination is not complete by 2 months. However, it is well established that Nav1.6 is a marker for mature nodes (Boiko et al., 2001, Kaplan et al., 2001) and we used immunocytochemistry and confocal microscopy to check that all the nodes we measured were positive for Nav1.6 staining. We have now discussed this issue in the manuscript (subsection “Variation of node length in the optic nerve and cortex”, second paragraph; Discussion, second paragraph) and have added the reference suggested by the reviewer.